# HOXB5 Overexpression Is Associated with Neuroendocrine Differentiation and Poor Prognosis in Prostate Cancer

**DOI:** 10.3390/biomedicines9080893

**Published:** 2021-07-26

**Authors:** Yohei Sekino, Quoc Thang Pham, Kohei Kobatake, Hiroyuki Kitano, Kenichiro Ikeda, Keisuke Goto, Shogo Inoue, Tetsutaro Hayashi, Masaki Shiota, Wataru Yasui, Jun Teishima

**Affiliations:** 1Department of Urology, Graduate School of Biomedical and Health Sciences, Hiroshima University, Hiroshima 734-8551, Japan; koukoba2710@gmail.com (K.K.); tanokin@hiroshima-u.ac.jp (H.K.); kenikeda@hiroshima-u.ac.jp (K.I.); keigoto@hiroshima-u.ac.jp (K.G.); inosyogo@hiroshima-u.ac.jp (S.I.); tetsu-haya@hiroshima-u.ac.jp (T.H.); teishima@hiroshima-u.ac.jp (J.T.); 2Department of Molecular Pathology, Graduate School of Biomedical and Health Sciences, Hiroshima University, Hiroshima 734-8551, Japan; quocthang388@gmail.com (Q.T.P.); wyasui@hiroshima-u.ac.jp (W.Y.); 3Department of Urology, Graduate School of Medical Sciences, Kyushu University, Fukuoka 812-8582, Japan; shiota.masaki.101@m.kyushu-u.ac.jp

**Keywords:** prostate cancer, HOXB5, RET, neuroendocrine differentiation, p53, RB1

## Abstract

Homeobox genes function as master regulatory transcription factors during embryogenesis. HOXB5 is known to play an important role in several cancers. However, the biological role of HOXB5 in prostate cancer (PCa) is not fully elucidated. This study aimed to analyze the expression and function of HOXB5 and involvement of HOXB5 in neuroendocrine differentiation in PCa. Immunohistochemistry showed that 56 (43.8%) of 128 cases of localized PCa were positive for HOXB5. HOXB5-positive cases were associated with poor prostate-specific antigen recurrence-free survival after prostatectomy. Among 74 cases of metastatic PCa, 43 (58.1%) were positive for HOXB5. HOXB5 expression was higher in metastatic PCa than that in localized PCa. HOXB5 knockdown suppressed cell growth and invasion, but HOXB5 overexpression increased cell growth and invasion in PCa cell lines. Furthermore, HOXB5 regulated RET expression. Gene set enrichment analysis revealed that Nelson androgen response gene set was enriched in low HOXB5 expression group. RB1 knockout increased HOXB5 expression. Of note, additional p53 knockdown further increased HOXB5 expression in RB1 knockout cells. In silico analysis showed that HOXB5 expression was increased in neuroendocrine PCa (NEPC). These results suggest that HOXB5 may be a promising prognostic marker after prostatectomy and is involved in progression to NEPC.

## 1. Introduction

Prostate cancer (PCa) is a leading cause of morbidity and mortality in many regions of the world [1]. Although androgen deprivation therapy (ADT) is initially effective, PCa finally progresses to castration-resistant PCa (CRPC), a life-threatening disease [2]. Although new antiandrogen drugs have recently provided significant benefits, the overall survival of patients with CRPC remains unsatisfactory [3]. Furthermore, neuroendocrine prostate cancer (NEPC) is a lethal subset of PCa that is increasingly occurring due to the increased use of antiandrogen drugs [4]. NEPC is castration-resistant and unaffected by antiandrogen drugs due to a lack of androgen receptor and associated signaling [5]. Thus, there is an urgent need to clarify the mechanisms of progression to CRPC and NEPC.

Homeobox (HOX) genes function as master regulatory transcription factors during embryonic development such as body patterning, growth, and differentiation [6]. The HOX family is classified into four chromosomal clusters (HOXA, HOXB, HOXC, and HOXD) [7]. Many recent studies have reported that dysregulation of the HOX family is involved in progression in some cancers including PCa [8]. Carriers of a mutation in the HOXB13 gene had a significantly increased risk of PCa compared with non-carriers [9]. HOXA13 overexpression promoted cell growth, migration, and invasion and was correlated with a poor prognosis in PCa [10]. In contrast, overexpression of HOXC6 suppressed cell growth in PCa [11]. Within the HOX family, HOXB5 is involved in the development of the enteric nervous system [12]. Abnormalities of HOXB5 function leads to Hirschsprung’s disease through the receptor tyrosine kinase RET [13]. A recent transcriptomic analysis showed that HOXB5 was increased in NEPC patient-derived organoids [14]. However, the expression and function of HOXB5 and involvement of HOXB5 in neuroendocrine differentiation (NED) in PCa has not been fully elucidated.

In this study, we analyzed the expression of HOXB5 in localized and metastatic PCa using whole-section immunohistochemistry. The effect of HOXB5 modulation on cell growth invasion and RET expression was evaluated in PCa cell lines. Furthermore, we examined the association between HOXB5 and NED in PCa.

## 2. Materials and Methods

### 2.1. Tissue Samples

In this retrospective study, 202 tumors were collected from patients diagnosed as having PCa who were treated between 2003 and 2014 at Hiroshima University Hospital (Hiroshima, Japan). This study was conducted in accordance with the Ethical Guidance for Human Genome/Gene Research of the Japanese Government. The Institutional Review Board of Hiroshima University Hospital approved this study (approval no. E-688).

We used archival formalin-fixed, paraffin-embedded tissues from the 202 patients with PCa for immunohistochemical analysis. Among these patients, 128 patients with clinically localized PCa were treated with radical prostatectomy (Appendix A). The cases with pathologically positive for resection margin were excluded. Tumor staging was performed according to the American Joint Committee on Cancer classification system [15]. Gleason grading was performed according to the international society of urological pathology (ISUP) consensus [16]. Biochemical relapse was defined as a prostate-specific antigen (PSA) level of ≥0.2 ng/mL [17]. Among the 202 patients, 74 patients with metastatic PCa were treated with ADT. PCa tissue samples were collected by biopsy for diagnostic examination in all 74 cases (Appendix A). Biochemical progression after ADT was defined according to the EAU-ESTRO-SIOG guidelines [18].

### 2.2. Immunohistochemistry

Immunohistochemistry was performed as described previously [19]. Sections were incubated with anti-HOXB5 antibody (1:100) (Thermo Fisher, Waltham, MA, USA) for 1 h at room temperature. HOXB5 expression in PCa was scored in all tumors as positive or negative. HOXB5 expression was scored in PCa and non-neoplastic prostate. When staining of HOXB5 was stronger in PCa than that in non-neoplastic prostate, the specimen was considered positive for HOXB5. Two observers (YT and WY) without knowledge of the patients’ clinical and pathologic parameters or outcomes independently reviewed immunoreactivity in each specimen.

### 2.3. In Silico Analysis

The GEPIA web tool was used to determine HOXB5 expression in The Cancer Genome Atlas (TCGA) (PRAD) dataset [20]. The expression array data were downloaded from GEO under accession numbers GSE3325 [21], GSE6099 [22], GSE21034 [23], GSE32269 [24], GSE32448 [25], GSE32967 [26], GSE35988 [27], GSE54460 [28], GSE66187 [29], GSE70770 [30], GSE77930 [31], GSE104786 [32], GSE126078 [33], and GSE147250 [34]. The Metastatic CRPC database was downloaded from the study by Abida et al. [35], and the NEPC database was downloaded from the study by Beltran et al. [36]. Gene set enrichment analysis (GSEA) was performed using the GSEA Java program (https://www.gsea-msigdb.org/gsea/index.jsp, accessed on 1 April 2021). Normalized enrichment score and FDR q values are shown in the figures. RET dependency data (DEMETER score) is based on pooled genome-scale shRNA screens from Project Achilles 2.201 [37].

### 2.4. Cell Lines

Three cell lines derived from human PCa (LNCaP: androgen-dependent cell line, DU145: androgen-independent cell line, and 22Rv-1: CRPC cell line) were kindly provided by Dr. Masaki Shiota (Kyushu University, Fukuoka, Japan). The cells were maintained in RPMI 1640 (Nissui Pharmaceutical Co., Ltd., Tokyo, Japan) containing 10% fetal bovine serum (BioWhittaker, Walkersville, MD, USA) in a humidified atmosphere with 5% CO_2_ at 37 °C.

### 2.5. Western Blotting

The lysates (40 µg) were solubilized in Laemmli sample buffer by boiling and subjected to 10% SDS-polyacrylamide gel electrophoresis followed by electro-transfer onto a nitrocellulose filter. The membrane was incubated with a primary antibody for HOXB5 (1:500) (Thermo Fisher, Waltham, MA, USA), p53(Do-1) (1:1000), RB1(1:1000), RET(1:1000) (Cell Signaling Technology, Inc., Danvers, MA, USA). Peroxidase-conjugated anti-mouse IgG or anti-rabbit IgG was used in the secondary reaction. Immunocomplexes were visualized with an ECL Western Blot Detection System (Amersham Biosciences, Piscataway, NJ, USA). β-actin (Sigma-Aldrich, St. Louis, MO, USA) was also stained as a loading control [1].

### 2.6. Generation of HOXB5-Expressing Cells

The coding sequence of HOXB5 was subcloned into the retroviral vector pDon-5 neo (Takara, Shiga, Japan), thus generating the HOXB5 overexpression vector. The coding sequence of HOXB5 and pDon-5 Neo were transfected into LNCaP cell lines by lipofection using Fugene-6 (Promega, Madison, WI, USA). After transfection, cells were released into drug-free medium for 48 h followed by G418 selection until single colonies were formed. Single clones were expanded, and overexpression of HOXB5 was confirmed by Western blotting.

### 2.7. RNA Interference

Silencer^®^ Select (Ambion, Austin, TX, USA) against HOXB5 and TP53 was used for RNA interference. Transfection was performed using Lipofectamine RNAiMAX (Invitrogen, Carlsbad, CA, USA).

### 2.8. Cell Growth and Invasion Assays

To examine cell viability, WST-1 assay was performed. The cells were seeded at a density of 2000 cells per well in 96-well plates. Cell growth was monitored after 1, 2, and 4 days. Three independent experiments were performed.

Modified Boyden chamber assays were performed to examine cell invasiveness. Cells were plated at 10,000 cells per well in RPMI 1640 medium in the upper chamber of a Transwell Insert (8-µm pore diameter; Chemicon, Temecula, CA, USA) coated with Matrigel. Medium containing 10% serum was added in the bottom chamber. After two days, cells in the upper chamber were removed by scraping, and the cells remaining on the lower surface of the insert were stained with CyQuant GR dye to assess the number of cells.

### 2.9. Generation of RB1 Knockout Cells

To knock out RB1 in LNCaP and 22Rv-1 cells, we used CRISPR-Cas9 technology, which was performed as described previously [38]. RB1 single-guide RNAs (CRISPR-RB1 vector) and scrambled single-guide RNAs (empty vector) were purchased from ABM Inc. (Richmond, BC, Canada). Lentiviral particles were generated by co-transfection of HEK293T cells with Cas9-sgRNA constructs and packaging plasmids (GAG, VSVG, REV). After 48 h, the conditioned media containing lentiviral particles were harvested and used to infect cells using polybrene as the transfection agent. Stable RB1 knockout cells were selected by passaging in media containing 4 µg/mL puromycin.

### 2.10. Enzalutamide and Apalutamide Treatment

Enzalutamide and apalutamide was obtained from Wako (Osaka, Japan) and handled according to the manufacturer’s recommendations. An WST-1 assay was performed 48 h after bicalutamide treatment as described previously. Drug sensitivity curves and IC_50_ values were calculated using GraphPad Prism 4.0 (GraphPad Software, La Jolla, CA, USA).

### 2.11. Statistical Analysis

All experiments were repeated at least three times with each sample in triplicate. The results are expressed as the mean ± SD of the triplicate measurements. Sample sizes for relevant experiments were determined by power analysis. Statistical differences were evaluated using the two-tailed Student *t*-test or Mann-Whitney *U*-test. A paired *T*-test was used to compare the statistical differences between PCa tissues and their corresponding non-neoplastic prostate tissues. The one-way analysis of variance (ANOVA) is used to determine whether there are any statistically significant differences between the means of 5 Gleason score groups. A *p*-value of <0.05 was considered statistically significant. After Kaplan-Meier analysis was performed, any statistical difference between the survival curves of the cohorts was determined with the log-rank Mantel-Cox test. Statistical analyses were conducted primarily using GraphPad Prism software (GraphPad Software Inc., La Jolla, CA, USA) or JMP14 (SAS Institute, Cary, NC, USA).

## 3. Results

### 3.1. Expression of HOXB5 in Localized PCa

We performed immunohistochemistry to analyze the expression of HOXB5 in 128 localized PCa tissue samples (Hiroshima cohort, Appendix A). Staining of HOXB was observed in the non-neoplastic prostate and PCa (Figure 1A,B). When staining of HOXB5 was stronger in PCa than that in non-neoplastic prostate, the specimen was considered positive for HOXB5. In total, 56 (43.8%) of the PCa cases were considered positive for HOXB5 (Figure 1B). The expression of HOXB5 was mainly detected in the cell cytoplasm of PCa (Figure 1C). The expression of *HOXB5* did not differ between non-neoplastic prostate and PCa in the GSE32448 data (Figure 1D). The positive cases for HOXB5 were associated with high Gleason score, high Gleason grade (ISUP), high T stage, and seminal vesicle invasion (Table 1). *HOXB5* expression was associated with high T stage, lymph node metastasis, and high Gleason score and in TCGA (PRAD) database (Figure 1E). A Kaplan–Meier analysis revealed that positive HOXB5 cases were associated with poor PSA recurrence-free survival (RFS) (*p* < 0.001, hazard ratio (HR) 6.823) (Figure 1F). The values of HOXB5 expression for PSA recurrence after radical prostatectomy were 70% sensitivity and 86% specificity. Similar results were observed in the findings from TCGA (PRAD), GSE21034, GSE54460, and GSE70770 (Figure 1G). We performed univariate and multivariate Cox proportional hazard analyses to evaluate the potential use of HOXB5 expression as a prognostic marker. In the univariate analysis, preoperative PSA, pT classification, Gleason score, and HOXB5 expression were associated with RFS. In the multivariate model, positive HOXB5 expression was independently associated with RFS (HR 6.032, *p* < 0.001; Table 2).

### 3.2. Expression of HOXB5 in Metastatic PCa

Then, we performed immunohistochemistry to analyze the expression of HOXB5 in the 74 metastatic PCa tissue samples (Hiroshima cohort, Appendix A). Among them, 43 (58.1%) cases were positive for HOXB5. The ratio of HOXB5 positive cases was higher in metastatic PCa than in localized PCa (Figure 2A), which was consistent with the findings from GSE3325 GSE6099, and GSE21034 (Figure 2B). A Kaplan–Meier analysis showed that the expression of HOXB5 was significantly associated with poor progression-free survival in patients with metastatic PCa treated with ADT (HR 2.221, *p* = 0.021) (Figure 2C). To examine whether HOXB5 was involved in progression to CRPC, we analyzed the expression of HOXB5 in CRPC patients using the public databases GSE32269, GSE35988, and GSE70770. The expression of *HOXB5* was higher in CRPC than that in treatment naïve PCa (Figure 2D).

### 3.3. HOXB5 Promotes Cell Growth and Invasion in PCa

The above findings suggested that HOXB5 may be involved in progression and metastasis in PCa. Then, we disrupted HOXB5 expression in PCa cell lines and investigated the effect of this modulation on cell growth and invasion. The expression of HOXB5 was higher in DU145 and 22Rv-1 cells than that in LNCaP cells (Figure 3A). We constructed pDON-5-HOXB5 expressing vector and transfected it into LNCaP cells. Western blotting showed that HOXB5 was increased in LNCaP cells transfected with the HOXB5 expression vector (Figure 3B). We examined the WST-1 and invasion assay and found that the overexpression of HOXB5 promoted cell growth and invasion (Figure 3C,D). Then, we used RNA interference targeting HOXB5 in DU145 and 22Rv-1 cells and confirmed the efficiency of HOXB5 knockdown by Western blotting (Figure 3E). HOXB5 knockdown suppressed cell growth and invasion (Figure 3F,G). These results suggested that HOXB5 is involved in cell growth and invasion in PCa cell lines.

### 3.4. Association between HOXB5 and RET in PCa

A recent paper showed that HOXB5 was expressed in enteric ganglia cells which also expressed high level of RET [12]. Of note, some studies have shown that HOXB5 binds to the promoter of the RET gene and enhances RET transcription [39,40]. Therefore, we analyzed the association between HOXB5 and RET in PCa. Western blotting showed that HOXB5 overexpression increased RET expression in LNCaP cells, and HOXB5 knockdown suppressed RET expression in 22Rv-1 cells (Figure 4A). In the study by Beltran et al. [36], gene alteration of *HOXB5* was associated with gene alteration of *RET* (Table 3). Furthermore, there was a moderate association between *HOXB5* expression and *RET* expression at the mRNA levels in CRPC public databases (GSE77930 and GSE126078) (Figure 4A). These results suggested that HOXB5 expression was correlated with RET expression at genomic, mRNA, and protein levels.

### 3.5. Involvement of HOXB5 in Neuroendocrine Differentiation in PCa

A recent study showed that RET expression and RET pathway activity were upregulated in NEPC patient samples [41]. Therefore, we analyzed the association between HOXB5 and NED in PCa. We conducted a GSEA analysis on the low HOXB5 expression group and high HOXB5 expression group focusing on a Nelson response to androgen gene set [42]. GSEA showed that the Nelson response to androgen gene set was enriched in the low HOXB5 expression group in TCGA (PRAD) and metastatic CRPC (Abida et al. [35]) (Figure 5A). What is more, we examined the dependency of HOXB5 on cell growth and viability using Project Achilles DEMETER scores in PCa cell lines [37]. DEMETER scores showed that the dependency on HOXB5 was greater in androgen receptor (AR)-negative PCa cell lines than that in AR-positive cell lines (Figure 5B). *HOXB5* expression was increased in NEPC from several public databases (GSE32967, GSE104786, GSE126078, and the study by Abida et al. [35]) (Figure 5C). Moreover, *HOXB5* expression was moderately correlated with NEPC score in the study by Abida et al. (Figure 5D). Of note, there was slight inverse correlation between *HOXB5* expression and PSA expression in the metastatic PCa public database GSE77930 (Figure 5E).

### 3.6. HOXB5 Expression Is Upregulated by RB1 and p53 Modulation

Recent studies have shown that p53 and RB1 are involved in cell plasticity and NED [34,43]. Therefore, we examined the association between HOXB5 and p53 and RB1. In metastatic CRPC public databases (Abida et al. [35]), *HOXB5* expression was increased in mutated *TP53* compared to that in wild type *TP53*. *HOXB5* expression was also increased in deleted *RB1* compared to that in diploid *RB1*. Of note, HOXB5 expression was increased in mutated TP53 and deleted RB1 compared to that in wild type *TP53* and diploid *RB1* (Figure 6A). We established RB1 knockout cells using the CRISPR-Cas9 system in LNCaP and 22Rv-1 cells because DU145 cells is RB1-deficient cell line. HOXB5 expression was upregulated in RB1 knockout cells. Furthermore, additional p53 knockdown using siRNA induced HOXB5 expression in LNCaP and 22Rv-1 cells (Figure 6B). To validate our finding, we used the public database (GSE147250), which is derived from double (*RB1* and *TP53*) knockout LNCaP cells. *HOXB5* expression was not significantly different between double knockout cells and control cells in LNCaP (Appendix A).

### 3.7. Involvement of HOXB5 in Enzalutamide and Apalutamide Resistance in PCa

The above findings indicate that HOXB5 was involved in NED. Therefore, we analyzed the involvement of HOXB5 in resistance to antiandrogen drugs (enzalutamide and apalutamide) in PCa cell lines. We performed WST-1 assays to measure cell viability under various concentrations of enzalutamide and apalutamide in LNCaP cells transfected with HOXB5 expressing vector and empty vector, and 22Rv-1 cells transfected with negative control siRNAs and HOXB5 siRNA. HOXB5 overexpression suppressed the sensitivity to enzalutamide and apalutamide in LNCaP cells (Figure 7A). HOXB5 inhibition increased the sensitivity to enzalutamide and apalutamide in 22Rv-1 cells (Figure 7B).

## 4. Discussion

Although HOXB5 modulated cell growth and invasion in PCa cell lines in the present study, the precise mechanism of how HOXB5 enhances cell growth and invasion is still unclear. There are some reports on the role of HOXB5 in cancer progression. In pancreatic cancer and lung cancer, HOXB5 promoted cell growth and invasion through β-catenin activation [44,45]. In breast cancer, HOXB5 enhanced cell growth and invasion partly through RET, ERBB2, EGFR, and ESR1 [46]. In head and neck squamous cell carcinoma, HOXB5 plays an oncogenic role through the EGFR/Akt/Wnt/b-catenin signaling axis [47]. These findings indicate that HOXB5 promotes cell growth and invasion through several key oncogenic molecules across various cancers. In the present study, HOXB5 modulated RET expression in PCa cell lines. A recent study showed that RET enhances cell growth and invasion in vitro and in vivo in PCa [48], which may help to explain the involvement of HOXB5 in cell growth and invasion in the PCa cell lines. Indeed, immunohistochemistry revealed that HOXB5 expression was upregulated in PCa with aggressive features, such as high Gleason score, high T stage, and metastatic PC in the Hiroshima cohort and public databases. So far, there has been no report on the drug targeting HOXB5. Considering that HOXB5 was increased in CRPC and NEPC and was involved in cell growth, invasion, and resistance to enzalutamide and apalutamide in PCa, HOXB5 may be a promising therapeutic target.

Several mutations, such as *TP53* and *RB1*, are identified as key molecules in the development of NEPC [34,43]. However, there remains a vital need to understand the molecular characteristics of NEPC. A recent study showed that HOXB5 was increased in NEPC patient-derived organoids [14]. In the present study, several public databases revealed that the expression of *HOXB5* was increased in NEPC. GSEA showed that Nelson androgen response gene set was enriched in low HOXB5 expression group in TCGA (PRAD) and a metastatic CRPC cohort. RB1 knockout and p53 knockdown upregulated HOXB5 expression in LNCaP and 22RV-1 cells. Furthermore, there was a close association between *HOXB5* and *RET*, which is involved in NED [41]. These results indicate that HOXB5 may play an essential role in NED in PCa. However, HOXB5 may not be a specific marker for NEPC because NEPC is known to be present in approximately 10–20% of CRPC patients [49]. In the present study, immunohistochemistry showed that HOXB5 expression was observed in 43.8% of localized PCa. Western blotting revealed that HOXB5 expression was detected in 22Rv-1 cells, which is a well-known AR-positive CRPC cell line [50]. Therefore, it remains to be determined how HOXB5 expression is increased during the transition from localized PC to a NEPC phenotype.

The regulation of HOXB5 has not been fully clarified. A recent study showed that HOXB5 was regulated by miR-221 in papillary thyroid carcinoma [51]. Another study has shown that promoter hypermethylation of HOXB5 was observed in ovarian cancer [52]. As mentioned above, in the present study, HOXB5 expression was regulated by RB1 and p53 in PCa cell lines. Furthermore, in silico analysis showed that high *HOXB5* expression was associated with deleted *RB1* and mutated *TP53*. A prior study reported that *TP53* inactivation by biallelic mutation or genomic copy loss occurs in 40–50% and that *RB1* inactivation by deletion occurs in about 10% of metastatic PCa and CRPC [53,54], indicating that the ratio of TP53 and RB1 alteration is higher in metastatic PCa than that in localized PCa. In the present study, immunohistochemistry and in silico analysis showed that HOXB5 expression was increased in metastatic PCa and CRPC compared to localized PCa. Although the precise mechanism for how p53 and RB1 regulate HOXB5 expression remains unclear, the network between HOXB5 and TP53 and RB1 may play a decisive role in progression to metastatic PCa, CRPC, and NEPC.

This study has some limitations. Although we focused on the involvement of HOXB5 in NED, we did not use NEPC cell lines. NCI-H660 is one such well-established NEPC cell line [41]. Besides, although we showed the close correlation between HOXB5 and RET, the mechanism how HOXB5 and RET involve in NED is unclear. Furthermore, although we showed that HOXB5 expression was increased in NEPC from the public databases, we did not analyze HOXB5 expression using clinical samples from NEPC. NEPC samples are difficult to collect because NEPC is a rare subset of CRPC.

## 5. Conclusions

In conclusion, the present study revealed that high expression of HOXB5 was associated with poor prognosis in localized and metastatic PCa. HOXB5 modulated cell growth and invasion partly through RET. HOXB5 expression was regulated by RB1 and p53 and was increased in NEPC. The data presented here highlight the great potential of HOXB5 as a potential biomarker and therapeutic target in PCa.

## Figures and Tables

**Figure 1 biomedicines-09-00893-f001:**
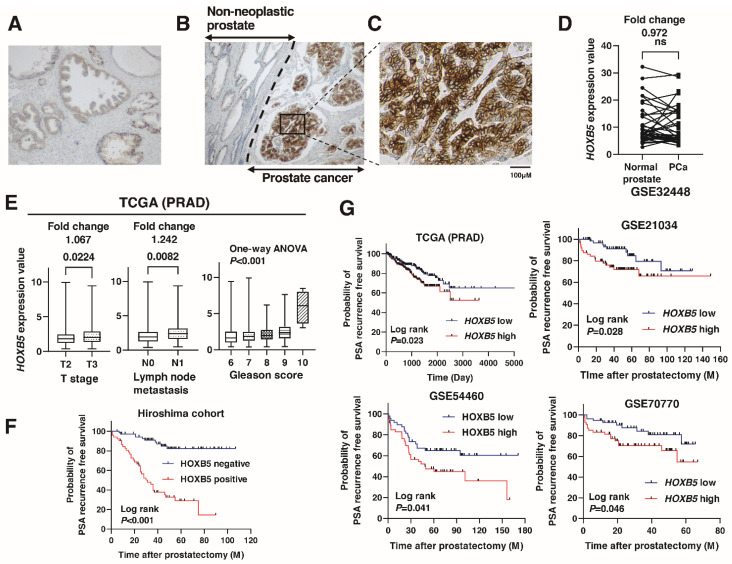
The expression of HOXB5 in localized prostate cancer (PCa). (**A**) Immunohistochemical staining in non-neoplastic prostate. Original magnification: 100×. (**B**) Immunohistochemical staining of HOXB5 in PCa. Original magnification: 100×. (**C**) Immunohistochemical staining of HOXB5 in PCa. Original magnification: 400×. Scale bar = 100 μm. (**D**) Scatter plot of HOXB5 expression in normal prostate and PCa from GSE32448. ns: not significant. Fold changes are indicated. (**E**) Comparisons of *HOXB5* mRNA expression levels between T stages, lymph node metastasis, and Gleason scores are displayed as a box plot from TCGA (PRAD) datasets. Fold changes are indicated. (**F**) Kaplan–Meier plot of PSA recurrence-free survival of PCa patients with HOXB5 expression after prostatectomy from the Hiroshima cohort. (**G**) Kaplan–Meier plot of PSA recurrence-free survival of PCa patients with HOXB5 expression after prostatectomy from TCGA (PRAD), GSE21034, GSE54460 and GSE70770.

**Figure 2 biomedicines-09-00893-f002:**
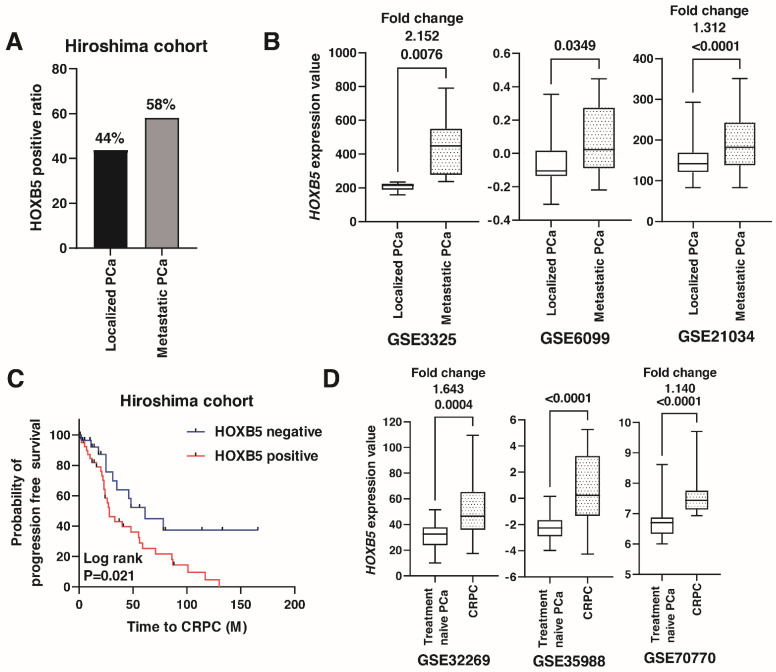
The expression of HOXB5 in metastatic prostate cancer (PCa). (**A**) The ratio of HOXB5-positive cases in localized and metastatic PCa from the Hiroshima cohort. (**B**) Box plot of HOXB5 expression in localized and metastatic PCa in GSE3325, GSE6099, and GSE21034. Fold changes are indicated. (**C**) Kaplan-Meier plot of progression-free survival of PCa patients with HOXB5 expression after androgen deprivation therapy from the Hiroshima cohort. (**D**) Box plot of HOXB5 expression in treatment naïve PCa and castration-resistant PCa (CRPC) in GSE32269, GSE35988, and GSE70770. Fold changes are indicated.

**Figure 3 biomedicines-09-00893-f003:**
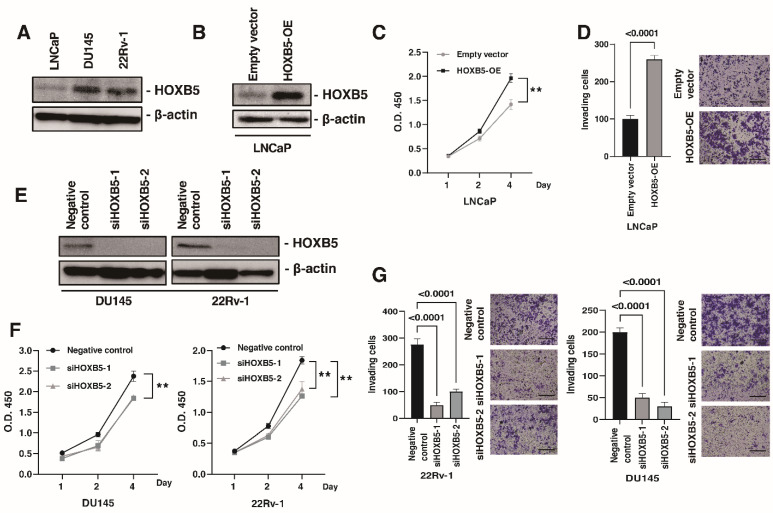
HOXB5 promotes cell growth and invasion in prostate cancer (PCa) cell lines. (**A**) Western blotting of HOXB5 in LNCaP, DU145, and 22RV-1 cells. β-actin was used as a loading control. (**B**) Western blotting of HOXB5 in LNCaP cells transfected with HOXB5 expressing vector and empty vector. β-actin was used as a loading control. (**C**) Cell growth assay in LNCaP cells transfected with HOXB5 expressing vector and empty vector. WST-1 assays assessed cell growth at 1, 2, and 4 days after seeding on 96-well plates. Bars and error bars indicate the mean and S.D., respectively, of three independent experiments. ** *p* < 0.01. (**D**) Cell invasion assay in LNCaP cells transfected with HOXB5 expressing vector and empty vector. Cell invasion was assessed using a modified Boyden chamber assay. Bars and error bars indicate the mean and S.D., respectively, of three independent experiments. Representative images of the invasion assay are shown. Scale bar, 100 μm. (**E**) Western blotting of HOXB5 in DU145 and 22Rv-1 cells transfected with HOXB5 or negative control siRNAs. β-actin was used as a loading control. (**F**) Cell growth assay in DU145 and 22Rv-1 cells transfected with HOXB5 or negative control siRNAs. WST-1 assays assessed cell growth at 1, 2, and 4 days after seeding on 96-well plates. Bars and error bars indicate the mean and S.D., respectively, of three independent experiments. ** *p* < 0.01. (**G**) Cell invasion assay in DU145 and 22Rv-1 cells transfected with HOXB5 or negative control siRNAs. Cell invasion was assessed using a modified Boyden chamber assay. Bars and error bars indicate the mean and S.D., respectively, of three independent experiments. Representative images of the invasion assay are shown. Scale bar, 100 μm.

**Figure 4 biomedicines-09-00893-f004:**
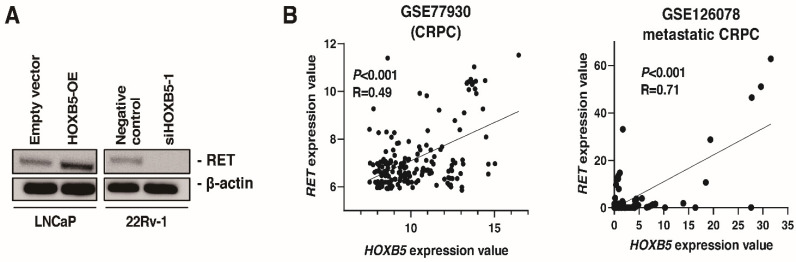
The association between HOXB5 and RET in prostate cancer (PCa). (**A**) Western blotting of RET in LNCaP cells transfected with HOXB5 expressing vector and empty vector and 22Rv-1 cells transfected with HOXB5 or negative control siRNAs. β-actin was used as a loading control. (**B**) The correlation between HOXB5 expression and RET expression in GSE77930 and GSE126078. Spearman’s correlation coefficients and *p*-values are indicated.

**Figure 5 biomedicines-09-00893-f005:**
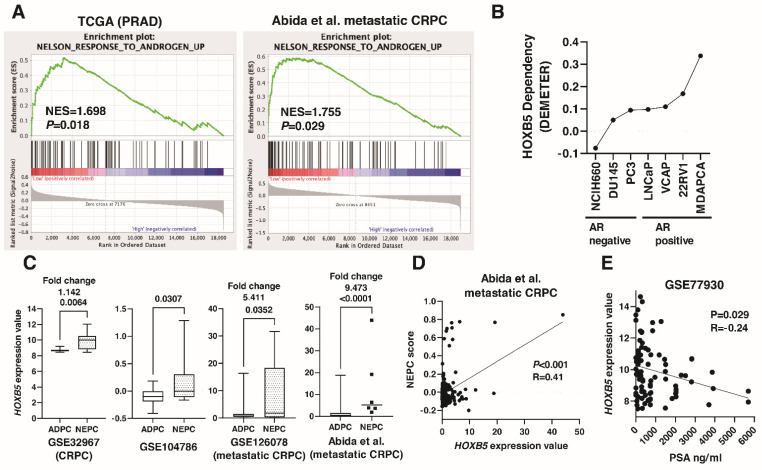
The involvement of HOXB5 in neuroendocrine differentiation in prostate cancer (PCa). (**A**) GSEA for genes differentially expressed in the low HOXB5 expression group compared to the high HOXB5 expression group using Nelson androgen response gene sets in TCGA (PRAD) and the study by Abida et al. (**B**) Relative HOXB5 dependency scores (DEMETER) reflect the ability of PCa cell lines to maintain proliferation after HOXB5 knockdown taken from Project Achilles 2.201. (**C**) Box plot of HOXB5 expression in ADPC and NEPC in GSE32967, GSE104786, GSE126078, and the study by Abida et al. Fold changes are indicated. (**D**) The correlation between HOXB5 expression and NEPC score in the study by Abida et al. Spearman’s correlation coefficients and *p*-values are indicated. (**E**) The correlation between HOXB5 expression and PSA expression in GSE77930. Spearman’s correlation coefficients and *p*-values are indicated.

**Figure 6 biomedicines-09-00893-f006:**
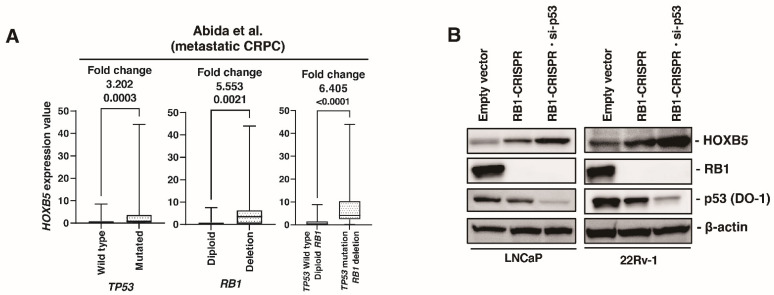
HOXB5 expression is upregulated in RB1 and p53 modulation. (**A**) Box plot of HOXB5 expression in wild type and mutated TP53, in diploid and deleted RB1, and TP53 wild type/diploid RB1 and TP53 mutation/RB1 deletion in the metastatic CRPC dataset (Abida et al.). Fold changes are indicated. (**B**) Western blotting of HOXB5, Rb1, and p53 (DO-1) in LNCaP and 22Rv-1 cells transfected with empty vector, RB1-CRISPR vector, RB1-CRISPR vector and si-p53. β-actin was used as a loading control.

**Figure 7 biomedicines-09-00893-f007:**
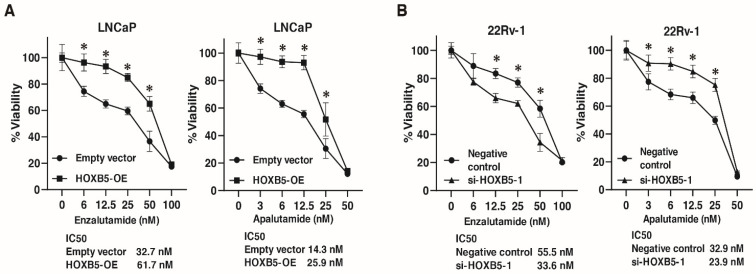
HOXB5 is involved in enzalutamide and apalutamide resistance in PCa. (**A**) The dose-dependent effects of enzalutamide and apalutamide on the viability of LNCaP cells transfected with HOXB5 expressing vector (HOXB-OE) and empty vector. * *p* < 0.05. IC50 values are indicated. (**B**) The dose-dependent effects of enzalutamide and apalutamide on the viability of 22Rv-1 cells transfected with negative control siRNAs and HOXB5 siRNA (si-HOXB5-1). * *p* < 0.05. IC50 values are indicated.

**Table 1 biomedicines-09-00893-t001:** Relationship between HOXB5 expression and clinicopathologic characteristics in the 128 patients with prostate cancer.

	HOXB5 Expression		*p*-Value ^a^
	Positive (*n* = 56) (%)	Negative (*n* = 72) (%)	
Age			
≤65 (*n* = 53)	23 (41%)	30 (42%)	N.S.
≥66 (*n* = 75)	33 (59%)	42 (58%)	
Preoperative PSA (ng/mL)			
≥10 (*n* = 78)	29 (52%)	49 (68%)	N.S.
<10 (*n* = 50)	27 (48%)	23 (31%)	
Gleason score			
6/7 (*n* = 84)	28 (50%)	56 (78%)	0.001
8/9/10 (*n* = 44)	28 (50%)	16 (22%)	
Gleason grade (ISUP)			
1/2 (*n* = 45)	12 (21%)	33 (46%)	0.001
3/4/5 (*n* = 83)	44 (79%)	39 (54%)	
pT stage			
pT2 (*n* = 105)	39 (70%)	66 (92%)	0.001
pT3 (*n* = 23)	17 (30%)	6 (8%)	
Seminal vesicle invasion			
Negative (*n* = 123)	51 (91%)	72 (100%)	0.003
Positive (*n* = 5)	5 (9%)	0 (0%)	

Abbreviations: PSA: prostate-specific antigen; N.S.: not significant. ^a^
*p* values were calculated with Fisher’s exact test.

**Table 2 biomedicines-09-00893-t002:** Univariate and multivariate Cox regression analysis of PSA recurrence-free survival.

	Univariate Analysis		Multivariate Analysis	
	HR (95% CI)	*p*-Value	HR (95% CI)	*p*-Value
Preoperative PSA (ng/mL)				
≤20	1 (Ref.)		1 (Ref.)	
>20	3.181 (1.348–7.506)	0.008	1.501 (0.6112–3.687)	0.3755
pT stage				
pT2	1 (Ref.)		1 (Ref.)	
pT3	3.137 (1.717–5.730)	<0.001	1.608 (0.855–3.028)	0.140
Gleason score				
6/7	1 (Ref.)		1 (Ref.)	
8/9/10	2.691 (1.533–4.723)	<0.001	1.806 (1.006–3.242)	0.048
HOXB5				
Negative	1 (Ref.)		1 (Ref.)	
Positive	7.802 (3.848–15.819)	<0.001	6.032 (2.876–12.648)	<0.001

Abbreviations: PSA: prostate-specific antigen; HR: hazard ratio; CI: confidence interval; Ref.: reference value.

**Table 3 biomedicines-09-00893-t003:** Relationship between HOXB5 and RET gene status in copy number alterations in the 107 metastatic prostate cancers in the study by Beltran et al. [36].

	*RET* Alteration		*p*-Value ^a^
	No (*n* = 66)	Yes (*n* = 41)	
*HOXB5* alteration			
No (*n* = 56)	51 (91%)	5 (9%)	<0.001
Yes (*n* = 51)	15 (29%)	36 (71%)	

^a^*p* values calculated with Fisher’s exact test.

## Data Availability

All data generated or analyzed during this study are included either in this article or in the Appendix A.

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
