# Peer review of "HOXB5 Overexpression Is Associated with Neuroendocrine Differentiation and Poor Prognosis in Prostate Cancer"

_biomedicines, 2021, doi:10.3390/biomedicines9080893_

Round 1
Reviewer 1 Report
Congratulations to the authors that have tried to improve the manuscript taking into account previous comments.
The manuscript has improved, however, the changes introduced have modified the redaction so it needs to be homogenies and reviewed.
Furthermore, there are some items that should be address:
- In Table 1, the authors have characterized the cohort of 128 patients according to two groups of Gleason score (6/7 and 8/9/10). It would be recommendable to include the ISUP grading system.
- The text format should be justified in the paragraphs “3.1. Expression of HOXB5 in Localized PCa”, “3.3. HOXB5 Promotes Cell Growth and Invasion in PCa”, among others.
- In Figure 2, the authors should include the Fold-Change value in the results from GSE6099 and GSE35988 datasets. This should be also corrected in Figure 5c.
- The authors have pointed out the association between HOXB5, RET, and NED. However, although they comment on this issue, this association should be further validated using in vitro approaches.
- It would be certainly interesting to validate the impact of double RB1 and TP53 loss on the expression of HOXB5 in available datasets such as the study of Abida et al.
Author Response
Our responses to the reviewers' comments for biomedicines-12499791
Title:HOXB5 overexpression is associated with neuroendocrine differentiation and poor prognosis in prostate cancer
Authors:Yohei Sekino, Quoc Thang Pham, Kohei Kobatake, Hiroyuki Kitano, Kenichiro Ikeda, Keisuke Goto, Shogo Inoue, Tetsutaro Hayashi, Masaki Shiota, Wataru Yasui, Jun Teishima
Dear Editor and Reviewers,
Thank you for the reviewers’ comments concerning our manuscript entitled “HOXB5 overexpression is associated with neuroendocrine differentiation and poor prognosis in prostate cancer”. Those comments are constructive and valuable for modifying and improving our manuscript. We have studied comments carefully and have fully addressed each comment. Revised portions are marked in red in the revised manuscript. I would like to resubmit this revised manuscript to biomedicines and hope it is acceptable for publication in the journal. The main corrections in the paper and the responses to the reviewer’s comments are as follows:
Reviewer 1
- In Table 1, the authors have characterized the cohort of 128 patients according to two groups of Gleason score (6/7 and 8/9/10). It would be recommendable to include the ISUP grading system.
Thank you for the valuable suggestion. We agree with the importance of ISUP grading system. Expectedly, the HOXB5 positive cases were associated with high Gleason grades.
These statements were described in the revised results (page3, line 138) and table 1.
- The text format should be justified in the paragraphs “3.1. Expression of HOXB5 in Localized PCa”, “3.3. HOXB5 Promotes Cell Growth and Invasion in PCa”, among others.
Thank you for the valuable comments. We justified text format in our manuscript.
- In Figure 2, the authors should include the Fold-Change value in the results from GSE6099 and GSE35988 datasets. This should be also corrected in Figure 5c.
Thank you for the valuable comments. Unfortunately, we could not calculate the fold changes in GSE6099 and GSE35988 because the data from GSE6099 and GSE35988 are already normalized and are negative values
- The authors have pointed out the association between HOXB5, RET, and NED. However, although they comment on this issue, this association should be further validated using in vitro approaches.
Thank you for the valuable suggestion. We totally agree with the reviewer’s comments. As we mentioned in the result of association between HOXB5 and RET in PCa, some study shown that HOXB5 binds to the promoter of the RET gene and enhances RET transcription in vitro and vivo, indicating that HOXB5 directly regulates RET expression. However, the involvement of HOXB5 and RET in neuroendocrine differentiation is still unclear. In the future, we will make RET knockout cells by CRISPR-Cas9 to examine the association between HOXB5, RET, and NED.
- It would be certainly interesting to validate the impact of double RB1 and TP53 loss on the expression of HOXB5 in available datasets such as the study of Abida et al.
Thank you for the valuable suggestion. We analyzed HOXB5 expression in mutated TP53 and deleted RB1 and found that HOXB5 expression was increased in mutated TP53 and deleted RB1.
These statements were described in the revised results (page10, line 273-274).
Once again, thank you very much for your comments and suggestions.
We are looking forward to hearing from you soon.
Yours Sincerely,
Yohei Sekino

Reviewer 2 Report
In the manuscript, Sekino et al have provided evidence showing that HOXB5 overexpression is associated with NED and poor prognosis in prostate cancer. Although this manuscript has some limitations, it was written concisely and their data were well presented.
Major concerns:
- Some citations were mismatched or missing: 1) In page 2, the citation for GSE32448 seems to be wrong (Ref. 24); 2) In Table 3, Beltran et al. [32] was an incorrect citation. Beltran et al. [35] or Labrecque et al. [32]?; 3) In page 8, Fig. 5, and Fig. 6, Abida et al. [31] is incorrect. Abida et al. [34] or Tsai et al. [31]?; 4) Reference to “the Nelson response to androgen gene set” was missing.
- In Fig. 5, is a NED signature changed in prostate cell lines when HOXB5 is overexpressed or its expression is inhibited?
- In Fig. 7, why don’t the authors use Rb1 knockout cell lines and Rb1 knockout plus p53 siRNA transfected cell lines for analysis of resistance to enzaluamide and apalutamide?
Minor concerns:
- In Fig. 1E, are HOXB5 expression levels statistically different between different Gleason scores?
- “~expressed in low HOXB5 expression group compared to the low HOXB5 expression group~” in Fig. 5 legend, the second “low” should be changed to “high”.
Author Response
Our responses to the reviewers' comments for biomedicines-12499791
Title:HOXB5 overexpression is associated with neuroendocrine differentiation and poor prognosis in prostate cancer
Authors:Yohei Sekino, Quoc Thang Pham, Kohei Kobatake, Hiroyuki Kitano, Kenichiro Ikeda, Keisuke Goto, Shogo Inoue, Tetsutaro Hayashi, Masaki Shiota, Wataru Yasui, Jun Teishima
Dear Editor and Reviewers,
Thank you for the reviewers’ comments concerning our manuscript entitled “HOXB5 overexpression is associated with neuroendocrine differentiation and poor prognosis in prostate cancer”. Those comments are constructive and valuable for modifying and improving our manuscript. We have studied comments carefully and have fully addressed each comment. Revised portions are marked in red in the revised manuscript. I would like to resubmit this revised manuscript to biomedicines and hope it is acceptable for publication in the journal. The main corrections in the paper and the responses to the reviewer’s comments are as follows:
Reviewer 2
- Some citations were mismatched or missing: 1) In page 2, the citation for GSE32448 seems to be wrong (Ref. 24); 2) In Table 3, Beltran et al. [32] was an incorrect citation. Beltran et al. [35] or Labrecque et al. [32]?; 3) In page 8, Fig. 5, and Fig. 6, Abida et al. [31] is incorrect. Abida et al. [34] or Tsai et al. [31]?; 4) Reference to “the Nelson response to androgen gene set” was missing.
We are deeply sorry for our mistakes. We carefully checked and revised our manuscript and added the appropriate citations to the Nelson response to androgen gene set.
- In Fig. 5, is a NED signature changed in prostate cell lines when HOXB5 is overexpressed or its expression is inhibited?
Thank you for the valuable suggestion. Although we showed that HOXB5 regulated RET expression in Prostate cancer cell lines, we did not show the effect of HOXB5 modulation on NED signature. Now, we are doing RNA sequence using HOXB5 overexpression cells and HOXB5 knockdown cells. In the future, we will surely present these results.
- In Fig. 7, why don’t the authors use Rb1 knockout cell lines and Rb1 knockout plus p53 siRNA transfected cell lines for analysis of resistance to enzaluamide and apalutamide?
Thank you for the valuable comments. We used prostate cancer cell lines transfected with HOXB5 overexpression vector or HOXB5 siRNAs to analyze the effect of HOXB5 modulation on anti-androgen drug resistance. Although we tried to analyze the effect of HOXB5 knockdown in RB1 knockout and p53 siRNA transfected cells, we failed to get successful results. The procedures using two siRNA (HOXB5 and TP53) are complicated. So, we speculate that they may affect the results. In the future, we will try to establish the double knockout (RB1 and TP53) cells and analyze the involvement of HOXB5 in anti-androgen drug resistance in double knockout cells.
Minor concerns:
- In Fig. 1E, are HOXB5 expression levels statistically different between different Gleason scores?
Thank you for the valuable suggestion. HOXB5 expression was increased according to Gleason score. One-way ANOVA test showed that HOXB5 expression was significantly different between different Gleason scores.
These statements were described in the revised figure 1E and statistical analysis (supplementary materials and methods.
- “~expressed in low HOXB5 expression group compared to the low HOXB5 expression group~” in Fig. 5 legend, the second “low” should be changed to “high”.
We are sorry for our mistake. We revise the figure legend in Figure 5.
These statements were described in the revised results (page9, line 259).
Once again, thank you very much for your comments and suggestions.
We are looking forward to hearing from you soon.
Yours Sincerely,
Yohei Sekino

Round 2
Reviewer 2 Report
Now this manuscript is acceptable.
This manuscript is a resubmission of an earlier submission. The following is a list of the peer review reports and author responses from that submission.
Round 1
Reviewer 1 Report
The discussions reflect their work in a plausible way.
There are some points that I would suggest.
It has been shown that HOXB5 is associated with tumor grade and progression in prostate cancer and other cancers. Is there any research on HOXB5-targeted therapy? In the Conclusion, it was mentioned that HOXB5 is a therapeutic target, so please describe HOXB5 as a therapeutic target as well.
Author Response
Dear Editor and Reviewers,
Thank you for the reviewers’ comments concerning our manuscript entitled “HOXB5 overexpression is associated with neuroendocrine differentiation and poor prognosis in prostate cancer”. Those comments are constructive and valuable for modifying and improving our manuscript. We have studied comments carefully and have fully addressed each comment. Revised portions are marked in red in the revised manuscript. I would like to resubmit this revised manuscript to biomedicines and hope it is acceptable for publication in the journal. The main corrections in the paper and the responses to the reviewer’s comments are as follows:
Reviewer 1
It has been shown that HOXB5 is associated with tumor grade and progression in prostate cancer and other cancers. Is there any research on HOXB5-targeted therapy? In the Conclusion, it was mentioned that HOXB5 is a therapeutic target, so please describe HOXB5 as a therapeutic target as well.
Thank you for the valuable comments. As far as we know, there has been no report on HOXB5-targeted therapy. Our study showed that HOXB5 was increased in CRPC and NEPC and was involved in cell growth and invasion. What is more, some studies have shown that HOXB5 affects several key oncogenic molecules, such as EGFR, ERBB2, AKT, and Wnt in cancer. These findings indicate that HOXB5 may be a promising therapeutic target in prostate cancer.
These statements were described in the revised discussion (page 10, line 300-302).
Once again, thank you very much for your comments and suggestions.
We are looking forward to hearing from you soon.
Yours Sincerely,
Yohei Sekino

Reviewer 2 Report
The authors have provided an interesting bird-eye view of the role of HOXB5 in the progression from primary PCa to castration-resistant (CRPC) and neuroendocrine-prostate cancer (NEPC). However, due to the general absence of in vitro and mechanistic studies of HOXB5 impact on neuroendocrine differentiation (NED), some comments should be considered by the authors.
The prostatectomy cohort rate of pt3 disease is lower than usually report, and for the proportion of patients with gleason 8 or more. Please explain in more detail and also the surgical margin, as the recurrence could be affected by this variable. Additionally, CRPC cohort should be further characterized (e.g. drug history).
Please define the descriptive statistic used in supplemental table.
The definition of expression of HOXB5 only based in immunohistochemistry is weak, and quantitative RNA expression should be performed to see fold change differences. Also, NEPC immunohistochemistry (e.g. NCAM1, CHGA, SYP) markers should be included into the analysis.
Different cell lines are used for different and similar experiments but no explanation about why each one is selected and if the experiments were also performed in the others cell lines with different results.
HOXB5 regulation pathways are supposed based on previous heterogeneous evidence as they are different in different types of tumour. A specific prostate cancer Microarray of Gene Expression Profile would have been very useful to evaluate its putative physio pathological role in this specific tumour.
The authors have pointed out the association between HOXB5, RET and NED. However, this association should be further validated using in vitro approaches
The authors have included GSE77930 study to further explore the expression levels of HOXB5 into NEPC samples. However, this study only considers localized and metastatic CRPC sample. For this reason, it should be excluded from the analysis.
Due to the reported interest of HOXB5 into neuroendocrine differentiation, it should be certainly interesting that authors analyse the correlation between HOXB5 expression and NEPC score reported by Abida et al. Furthermore, it would be highly recommended to evaluate the role of HOXB5 into NEPC-like cell lines (e.g. NCIH660 or PC-3 cells), as highlighted by the authors.
Considered the promising role of HOXB5 into ADT resistance, it would be interesting to analyse the effect of HOXB5 up- and down-regulation into ADT response in prostate cells.
The authors have explored the impact of RB1-loss and TP53 knockdown on the expression of HOXB5. Nevertheless, due to the absence of NEPC-like cell lines within this study (e.g. NCIH660 or PC-3 cells), it would be recommended to further characterize the role of HOXB5 within this NEPC-like model.
Author Response
Dear Editor and Reviewers,
Thank you for the reviewers’ comments concerning our manuscript entitled “HOXB5 overexpression is associated with neuroendocrine differentiation and poor prognosis in prostate cancer”. Those comments are constructive and valuable for modifying and improving our manuscript. We have studied comments carefully and have fully addressed each comment. Revised portions are marked in red in the revised manuscript. I would like to resubmit this revised manuscript to biomedicines and hope it is acceptable for publication in the journal. The main corrections in the paper and the responses to the reviewer’s comments are as follows:
Reviewer 2
- The prostatectomy cohort rate of pT3 disease is lower than usually report, and for the proportion of patients with gleason 8 or more. Please explain in more detail and also the surgical margin, as the recurrence could be affected by this variable. Additionally, CRPC cohort should be further characterized (e.g. drug history).
Thank you for the valuable comments on our cohort. Regarding the prostatectomy cohort, we excluded the cases with pathologically positive for resection margin because resection margin greatly affects the prognosis after prostatectomy. Although the reviewer mentioned CRPC cohort, we did not use CRPC cohort, but metastatic prostate cancer cohort.
These statements were described in the materials and methods (page 2, line 72-73).
- Please define the descriptive statistic used in supplemental table.
Thank you for your valuable suggestion. We revised supplementary table 1 and 2. We added the values of mean ± SD and some variables (D’Amico classification, PSA recurrence, and Time to CRPC).
These statements were described in the revised supplementary table 1 and 2.
- The definition of expression of HOXB5 only based in immunohistochemistry is weak, and quantitative RNA expression should be performed to see fold change differences. Also, NEPC immunohistochemistry (e.g. NCAM1, CHGA, SYP) markers should be included into the analysis.
Thank you for the critical comments.
#1 the definition of HOXB5 expression in immunohistochemical analysis
We revised the description of immunohistochemistry in the material and methods.
#2 the fold change differences
We calculated it and revised figures. Unfortunately, we could not calculate it in some figures because the data from some GSE are already normalized and are negative values
#3 NEPC immunohistochemistry
We did not perform immunohistochemistry using NEPC markers. But, now we are collecting NEPC samples and will show HOXB5 expression in NEPC samples. We revised the limitation.
These statements were described in the revised materials and methods (page 2, line 84-86), figures 1D, E, figure 2B, D, figure 5C, figure 6A and discussion (page 11, line 336-339)
- Different cell lines are used for different and similar experiments, but no explanation about why each one is selected and if the experiments were also performed in the others cell lines with different results.
Thank you for the valuable comments. We added some description of cell lines in materials methods. We excluded DU145 cells for making RB1 knockout cells because DU145 cells is RB1-deficient cell line. We also excluded DU145 cells for analyzing RET expression because RET expression is low in DU145 cells (data not shown).
These statements were described in the revised materials methods (page 3, line 102-103) and results (page 9, line 266-267).
- HOXB5 regulation pathways are supposed based on previous heterogeneous evidence as they are different in different types of tumour. A specific prostate cancer Microarray of Gene Expression Profile would have been very useful to evaluate its putative physio pathological role in this specific tumour.
Thank you for the valuable advice. As far as we know, we could not find the public database related to HOXB5 regulation pathways. But, we found the public cohort (GSE147250) using RB1 and TP53 knockout cells in LNCaP cells. Unfortunately, HOXB5 expression was not significantly different between double knockout cells and control cells in LNCaP. We added the supplementary figure 1 and revised the result of figure 6.
These statements were described in the revised result (page 9, line 269-272).
- The authors have pointed out the association between HOXB5, RET, and NED. However, this association should be further validated using in vitro approaches.
Thank you for the valuable comments. Although our study showed the close relationship between HOXB5 and RET, the mechanism how HOXB5 and RET involve in NED is unclear. In the future, we will make RET knockout cells by CRISPR to further examine the role of HOXB5 in NED. We revised the limitation.
These statements were described in the revised result (page 11, line 335-336).
- The authors have included GSE77930 study to further explore the expression levels of HOXB5 into NEPC samples. However, this study only considers localized and metastatic CRPC sample. For this reason, it should be excluded from the analysis.
Thank you again for the valuable advice. GSE77930 provides us the information (CHGA and SYP expression). Therefore, we considered CHGA and SYP positive cases as NEPC samples. But, we agree with the reviewer's opinion and excluded the result from GSE77930.
⑧ Due to the reported interest of HOXB5 into neuroendocrine differentiation, it should be certainly interesting that authors analyze the correlation between HOXB5 expression and NEPC score reported by Abida et al. Furthermore, it would be highly recommended to evaluate the role of HOXB5 into NEPC-like cell lines (e.g. NCIH660 or PC-3 cells), as highlighted by the authors.
#1 NEPC score
Thank you for the valuable suggestion. We examined the association between HOXB5 expression and NEPC score from the study (Abida et al.) and found a moderate correlation between them.
#2 NEPC-like cell lines
Although I tried to analyze the role of HOXB5 using NCIH660 cells, I failed to do so. As you know, NCIH660 cells is difficult to study because NCI-H660 cells grow slowly in vitro. Our colleagues at Kyoto University in Japan have established a unique NEPC cell line (KUCaP13). In the future, we will analyze the role of HOXB5 using KUCaP13.
These statements were described in the revised result (page 8, line 245-246).
⑨ Considered the promising role of HOXB5 into ADT resistance, it would be interesting to analyze the effect of HOXB5 up- and down-regulation into ADT response in prostate cells.
Thank you for the excellent suggestion. I have already analyzed the role of HOXB5 in enzalutamide resistance and found that HOXB5 expression was increased in enzalutamide-resistant cell lines. In the future, we will present these novel findings.
- The authors have explored the impact of RB1-loss and TP53knockdown on the expression of HOXB5. Nevertheless, due to the absence of NEPC-like cell lines within this study (e.g. NCIH660 or PC-3 cells), it would be recommended to further characterize the role of HOXB5 within this NEPC-like model.
Thank you for the valuable suggestion. Now, we are trying to generate double (RB1 and TP53) knockout cells using CRISPR. As we mentioned in question#5, our findings were not consistent with the results from the public database. We will validate our findings using double knockout cells and further examine the role of HOXB5 in NED.
Once again, thank you very much for your comments and suggestions.
We are looking forward to hearing from you soon.
Yours Sincerely,
Yohei Sekino

Round 2
Reviewer 1 Report
It has been modified according to the comments.
Reviewer 2 Report
This reviewer thanks the authors for the comment´s responses, however, most of the main concerns have not been address so the paper has not been improved from a scientific point of view. From my point of view, all the results that have been explored by the authors should be included and the available experiments performed before the paper could be published.